# Sulfated Hyaluronan Binds to Heparanase and Blocks Its Enzymatic and Cellular Actions in Carcinoma Cells

**DOI:** 10.3390/ijms23095055

**Published:** 2022-05-02

**Authors:** Jia Shi, Riku Kanoya, Yurina Tani, Sodai Ishikawa, Rino Maeda, Sana Suzuki, Fumiya Kawanami, Naoko Miyagawa, Katsuhiko Takahashi, Teruaki Oku, Ami Yamamoto, Kaori Fukuzawa, Motowo Nakajima, Tatsuro Irimura, Nobuaki Higashi

**Affiliations:** 1Department of Biochemistry, Hoshi University School of Pharmacy, 2-4-41, Ebara, Shinagawa-ku, Tokyo 144-8501, Japan; d2181@hoshi.ac.jp (J.S.); m2106@hoshi.ac.jp (R.K.); s171147@hoshi.ac.jp (Y.T.); s171012@hoshi.ac.jp (S.I.); s161238@hoshi.ac.jp (R.M.); 3737.feb@gmail.com (S.S.); s161095@hoshi.ac.jp (F.K.); s161252@hoshi.ac.jp (N.M.); ka-takahashi@hoshi.ac.jp (K.T.); 2Department of Microbiology, Hoshi University School of Pharmacy, 2-4-41, Ebara, Shinagawa-ku, Tokyo 144-8501, Japan; oku@hoshi.ac.jp; 3Department of Physical Chemistry, Hoshi University School of Pharmacy, 2-4-41, Ebara, Shinagawa-ku, Tokyo 144-8501, Japan; s161273@hoshi.ac.jp (A.Y.); k-fukuzawa@hoshi.ac.jp (K.F.); 4SBI Pharmaceuticals Co., Ltd., 1-6-1, Roppongi, Minato-ku, Tokyo 106-6019, Japan; motnakaj@sbigroup.co.jp; 5Division of Glycobiologics, Intractable Disease Research Center, Juntendo University School of Medicine, 2-1-1, Hongo, Bunkyo-ku, Tokyo 104-8520, Japan; t-irimura@juntendo.ac.jp

**Keywords:** heparanase, heparin, sulfated hyaluronan, heparan sulfate degradation, cell invasion in 3D culture, NF-κB, chemokine, fragment molecular orbital method

## Abstract

We examined whether sulfated hyaluronan exerts inhibitory effects on enzymatic and biological actions of heparanase, a sole endo-beta-glucuronidase implicated in cancer malignancy and inflammation. Degradation of heparan sulfate by human and mouse heparanase was inhibited by sulfated hyaluronan. In particular, high-sulfated hyaluronan modified with approximately 2.5 sulfate groups per disaccharide unit effectively inhibited the enzymatic activity at a lower concentration than heparin. Human and mouse heparanase bound to immobilized sulfated hyaluronan. Invasion of heparanase-positive colon-26 cells and 4T1 cells under 3D culture conditions was significantly suppressed in the presence of high-sulfated hyaluronan. Heparanase-induced release of CCL2 from colon-26 cells was suppressed in the presence of sulfated hyaluronan via blocking of cell surface binding and subsequent intracellular NF-κB-dependent signaling. The inhibitory effect of sulfated hyaluronan is likely due to competitive binding to the heparanase molecule, which antagonizes the heparanase-substrate interaction. Fragment molecular orbital calculation revealed a strong binding of sulfated hyaluronan tetrasaccharide to the heparanase molecule based on electrostatic interactions, particularly characterized by interactions of (−1)- and (−2)-positioned sulfated sugar residues with basic amino acid residues composing the heparin-binding domain-1 of heparanase. These results propose a relevance for sulfated hyaluronan in the blocking of heparanase-mediated enzymatic and cellular actions.

## 1. Introduction

Heparanase (Hpse) is a sole mammalian enzyme that cleaves heparan sulfate (HS) and macromolecular heparin into 5 to 10 kDa fragments. Hpse is essential for the invasion of cancer cells through the basement membrane by degrading HS proteoglycan in the extracellular matrix [1,2,3] and on the cell surface [4], also conceivably involved in the extravasation of circulating immune cells under inflammatory conditions [5,6,7]. These involvements are likely due to its enzymatic actions because the administration of Hpse enzyme inhibitors decreased the number of invasive cells [2,3,7]. In addition to the enzyme-dependent functions, Hpse stimulates many types of cells to induce cell adhesion [8], migration [9], and production of inflammatory cytokines and chemokines in an enzyme-independent manner [10,11,12,13,14]. It is suggested that clustering of cell surface HS on the target cells is essential to signal transduction [15]. Regarding intracellular signaling of Hpse after binding to the cells, Akt- and ERK-dependent pathway [9,12], or TLR-2, TLR-4, and a downstream NF-κB-dependent pathway [10,11,13] are involved depending on the cell types tested.

In order to block the enzyme-dependent and -independent actions of Hpse involved in disease progression, numerous chemical products have been designed to act as Hpse inhibitors [16], including carbohydrates and carbohydrate mimetics. Fragmented heparin that lacks cleavable glucuronic acid residues and chemically modified heparin inhibit the Hpse-mediated degradation of HS via its competitive binding to Hpse [17,18]. Synthesized or chemically-modified saccharides, such as PI-88 (muparfostat), laminarin sulfate, PG545 (pixatimod), SST0001 (roneparstat), and lambda-carrageenan, have been reported as Hpse inhibitors [2,3,19,20,21]. These poly- or oligo-saccharides commonly possess sulfate groups, however, their carbohydrate backbones are different from HS. Therefore, our research is aimed at the identification of an optimized carbohydrate backbone other than HS/heparin that can efficiently suppress Hpse-mediated actions. In a previous paper, we focused on the interaction of Hpse with chondroitin sulfate (CS), including a highly sulfated CS (squid cartilage-derived chondroitin sulfate-E (CS-E)) and mouse bone marrow mast cell-derived glycosaminoglycan (GAG), and identified CS-E as a potential intrinsic inhibitor of Hpse [22]. Because CS-E lacks *N*-sulfation, our results implied that sulfated polysaccharides without *N*-sulfation can also be inhibitory to Hpse if the carbohydrates are properly sulfated.

Hyaluronan (HA) consists of repetitive disaccharide units of D-glucuronic acid and *N*-acetyl-D-glucosamine, which is similar to HS in carbohydrate composition. However, HA and HS differ in their linkage between two saccharides, that is, the backbone of HA is (GlcNAc β1-4 GlcA β1-3)-linked, whereas that of HS/heparin is (GlcNAc α1-4 GlcA β1-4)-linked. HA is not sulfated in nature, has a large molecular size (around 10^6^~10^7^ Da), and confers anti-inflammatory properties on macrophages via a receptor-mediated interaction [23]. The effect of the molecule depends on its molecular size, that is, fragmented HA as a product of chemical or enzymatic degradation acts as a damage-associated molecular pattern that transduces an inflammatory signal, at least partly due to a TLR4-dependent signal [24]. Overproduction of HA influences malignancy and invasiveness of cancer, which is partly due to reduced contact inhibition and enhancement of CD44-mediated signaling [25].

Chemical modification of HA has been attempted for the improvement of GAG-mediated functions. It has been reported that sulfated HA, one such HA derivative, has broad anti-inflammatory effects, such as suppression of lipopolysaccharide-induced NF-κB activation and induction of cytokine production in macrophages [26,27,28,29,30,31,32,33]. As a result, the administration of sulfated HA exerts therapeutic effects on inflammation-related diseases and symptoms, such as cationic cathelicidin peptide-induced cutaneous inflammation [27], periodontitis [29], skin wound healing [31,32], and radiation-induced oral mucositis [33]. In addition to their anti-inflammatory effects, sulfated HA derivatives are also reported to foster pro-osteogenic conditions by modulating extracellular matrix environments and by scavenging Wnt antagonists [34], to induce apoptosis in bladder cell carcinoma cells [35], to maintain the undifferentiated state and pluripotency of human-induced pluripotent stem cells in culture [36], to negatively regulate VEGF-A-mediated sprouting of endothelial cells [37], and to suppress bacterial infection via blocking of cell surface interactions with epithelial cells [38]. 

Considering the critical involvement of sulfate groups in carbohydrate-based Hpse inhibitors [2,3,17,18,19,20,21], we hypothesized that sulfated HA is inhibitory to Hpse-mediated enzymatic and non-enzymatic actions. In the present study, we examine the inhibitory effects of sulfated HA on Hpse-mediated HS degradation and Hpse-induced cellular responses. As a novel approach to understanding the molecular basis of the interaction between Hpse and sulfated HA, we performed a fragmented molecular orbital (FMO) calculation [39,40,41] based on the structural information of the Hpse-heparin tetrasaccharide (Dp4) interaction [42].

## 2. Results

### 2.1. Molecular Nature of the Sulfated HA Samples

The sulfation ratio, that is, the number of sulfate groups per disaccharide, of sulfated HA and commercially available heparin used in the current study was determined using a conductimetric method as described [43]. Heparin, low-sulfated HA (ls-HA), and high-sulfated HA (hs-HA), used as sulfated GAGs, contained 2.81 ± 0.43, 1.31 ± 0.10, and 2.53 ± 0.32 sulfate groups per disaccharide unit (mean ± S.D., *n* = 3), respectively. The molecular weights of heparin, ls-HA, and hs-HA were estimated to be around 15 kDa, 50~60 kDa, and 40~50 kDa, respectively.

### 2.2. Sulfated HA Inhibited Human and Mouse Hpse-Mediated Degradation of HS

The inhibitory effects of sulfated HA on Hpse-mediated HS degradation were examined. In the presence of ls-HA or hs-HA, HS degradation by human and mouse Hpse was suppressed in a similar manner by heparin (Figure 1, Appendix A). Non-sulfated HA with a molecular weight of 20~30 kDa (NaHA-T2) did not show any detectable inhibition of muHpse-mediated HS degradation (Appendix A). 

The inhibition efficiency was quantified as IC_50_ calculated from the percentage of relative degradation from three independent experiments. IC_50_ of hs-HA was significantly smaller than that of heparin (Table 1).

### 2.3. Immobilized Sulfated HA Bound to Human and Mouse Hpse Proteins

Different amounts of biotinylated GAGs were immobilized on a streptavidin-coated plate. Thereafter, the binding of hu- and mu-Hpse to the immobilized GAGs was examined using an enzyme-linked immunosorbent assay (ELISA). When the amounts of immobilized GAGs were serially diluted, hu- and mu-Hpse bound to hs-HA at lower immobilized amounts (Figure 2). 

The interaction of Hpse with heparin and hs-HA was further analyzed using surface plasmon resonance. MuHpse(mature) with a concentration ranging from 0.41 to 3.70 nM was measured as an analyte. The sensorgram showed substantial interaction of muHpse(mature) with the immobilized GAGs (Figure 3). Association and dissociation rate constants (k_a_ and k_d_), as well as dissociation equilibrium constants (K_D_), were determined in triplicates (Table 2). The difference in k_d_ (*p* = 0.0003) and K_D_ (*p* = 0.002) was significant, whereas that in k_a_ was not (*p* = 0.85).

### 2.4. Sulfated HA Inhibited Hpse-Mediated Invasion of Colon-26 and 4T1 Cells into a Collagen Gel

The inhibitory effects of heparin and sulfated HA on cellular invasion were examined using a 3D culture system [44]. Colon-26 A1-3 cells stably expressing muHpse transgene were highly invasive when the cell cluster was embedded in a collagen gel. The extent of the invasion was quantified by measuring the length of the cell extension protruding from the cell cluster body. The mean length of the cell extension was shorter when hs-HA was added to the collagen gel, indicating that hs-HA suppressed cellular invasion into the collagen gel. The expression level of another possible target for hs-HA, that is, hyaluronidase-1, was not altered by muHpse transgene expression (data not shown). hs-HA was similarly effective in suppressing cell extension of murine 4T1 cells (Figure 4).

### 2.5. Sulfated HA Inhibited Hpse-Mediated CCL2 Release from Colon-26 Cells

The inhibitory effects of heparin and sulfated HA on a proinflammatory effect of Hpse, that is, muHpse(pro)-mediated release of the chemokine CCL2 from colon-26 cells [14], were examined. The release of CCL2 was significantly lower in the presence of heparin or sulfated HA (Figure 5A). The binding of muHpse(pro) to the cell layer was mostly blocked in the presence of heparin or sulfated HA (Figure 5B). The muHpse(pro)-mediated release of CCL2 was inhibited by pretreatment with BAY11-7082, an inhibitor of κB kinase (Figure 5C). Treatment of the cells with sulfated HA but not unmodified HA reduced muHpse(pro)-mediated NF-κB-dependent luciferase expression (Figure 5D).

### 2.6. Fragmented Molecular Orbital Calculation of the Interaction between Sulfated GAG Tetrasaccharides and huHpse

The molecular interaction of sulfated GAG tetrasaccharides with huHpse was further analyzed by fragment molecular orbital (FMO) calculation [39,40,41]. To make the overall electrostatic condition in carbohydrates equal, sulfated HA tetrasaccharide with six sulfate groups (∆HexUA(2S)-GlcNAc4S(6S)-GlcUA(2S)-GlcNAc4S(6S), charged to -8*e:* hs-HA4) and heparin tetrasaccharide with six sulfate groups (∆HexUA(2S)-GlcNS(6S)-IdoUA(2S)-GlcNS(6S), charged to -8*e*: Dp4) were examined in the modeling study (Appendix A). Here, sulfation modification was defined as two sulfate groups in an *N*-acetyl-D-glucosamine residue and one group in a hexuronic or glucuronic acid residue.

Conformation of the tetrasaccharides that bind to huHpse was determined (Figure 6). The distance from (+1) to (−3) sugar residues was 10.87 Å (hs-HA4) and 9.43 Å (Dp4), respectively. The difference in the distances reflects the bending of the tetrasaccharide Dp4, as shown in the superimposed images shown in Figure 6C. As a result of the bending inside the Hpse molecule, Dp4 can fit deeper into the active site cleft of the Hpse molecule than hs-HA4.

To predict amino acid residues involved in hs-HA-mediated inhibition of Hpse, statistically corrected inter-fragment interaction energy (SCIFIE) [45] of Hpse and the tetrasaccharides was calculated (Appendix A). As a result, the overall SCIFIE was smaller, that is, the interaction was more stable, in Dp4 than in hs-HA4. The SCIFIE was dissected into each amino acid residue of Hpse (Figure 7). hs-HA4 (panel A) and Dp4 (panel B) showed similar SCIFIE patterns at each amino acid residue, that is, lower interaction energy was shown at Lys159, Lys161, Lys231, Lys232, Arg272, His296, Arg303, and Lys98. This suggests that these basic amino acid residues are commonly involved in stable interactions with the tetrasaccharides. The SCIFIE of Dp4 was subtracted from that of hs-HA4 (panels C, D), showing that Lys159, Lys161, and Lys98 in particular are involved in a relatively more stable interaction in hs-HA4.

## 3. Discussion

The physiological relevance of Hpse has been extensively examined for many pathological and homeostatic phenomena, but its interaction with carbohydrates other than HS or heparin has not attracted much attention. In a previous study, we identified CS-E as a potential intrinsic inhibitor of Hpse [22], which implies that carbohydrates with a backbone other than HS/heparin can suppress Hpse-mediated actions if they are properly sulfated. In the present study, we extended this concept to examine the inhibitory effects of sulfated HAs on Hpse-mediated actions. Sulfated HAs inhibited the enzymatic activity of human and mouse Hpse. hs-HA exerted an around 10 times stronger inhibitory effect than heparin. Binding tests and surface plasmon resonance partly explained the inhibitory effects of hs-HA. Sulfated HA had the capacity to bind human and murine Hpse at lower immobilized amounts. As a result of surface plasmon resonance analysis, hs-HA showed a stronger interaction with muHpse as evidenced by a lower K_D_ value mainly due to the difference in dissociation rate constants (Table 2). We confirmed that the inhibition was also effective on the cellular level. The addition of hs-HA significantly suppressed the cell extension from cancer cell clusters cultured in a collagen gel. The addition of sulfated HA also reduced muHpse-dependent CCL2 expression via blocking of Hpse cell surface binding and preventing subsequent NF-κB signaling. Extension of invading cells reflects the enzymatic function of Hpse, whereas CCL2 production reflects the non-enzymatic action of Hpse as a cytokine-like stimulator. The inhibitory action of hs-HA was structurally supported by molecular modeling, that is, hs-HA4 strongly interacts with Lys159, Lys161, and Lys98 of huHpse in an electrostatic manner. To our knowledge, the current study is the first report to show the inhibitory action of sulfated HA on Hpse-mediated enzymatic and cellular events.

Sulfated HAs have been identified as anti-inflammatory compounds that suppress the production of inflammatory cytokines/chemokines. It was reported that the administration of sulfated HA compounds was effective to suppress inflammation in several inflammatory disease models, such as cationic cathelicidin peptide-induced Rosacea [27], periodontitis [29], imiquimod-induced cutaneous inflammation [32], and radiation-induced oral mucositis [33]. Administration of sulfated HA as a component of artificial extracellular matrix together with type I collagen facilitates accelerated dermal wound healing in diabetic mice, at least partly via reducing inflammatory macrophage activity. Despite the anti-inflammatory and regenerative properties of sulfated HA, the target molecules involved in the biological events have not been fully uncovered, except for a few potential target molecules, such as cathelicidin [27], HB-EGF [31], sclerostin [34], and hyaluronidase-1 [35]. This is partly because sulfated HA accepts a relatively wide spectrum of target molecules. Sulfated HA-dependent downstream changes in signal transduction include decreased phosphorylation of NF-κB and upregulation of SOD2 and SOD3 expression as a defense system against reactive oxygen species [30]. In the present study, we demonstrated that Hpse, a critical factor for cancer metastasis and inflammation, acts as a target molecule for sulfated HA. We also showed that the inhibitory effect of sulfated HA, especially hs-HA, exceeded that of a conventional Hpse inhibitor, heparin, which is due to the stronger binding capacity to Hpse as shown in the binding test (Figure 2 and Figure 3). Unexpectedly, the sulfation ratio was slightly higher in heparin than in hs-HA as determined by a conductimetric method. Therefore, it is suggested that the position of sulfate groups in the GAG molecule or the linkage of sugar residues is responsible for the stronger interaction of hs-HA with Hpse. The position of sulfate groups is likely to influence possible side effects. Recently, it was reported that the sulfated HA had weaker blood anticoagulant effects than heparin [46]. It is possible that lack of 2-*N*-sulfation influences the lower anticoagulant effects as has been demonstrated in heparin derivatives [47].

Our study showed that hs-HA effectively inhibited Hpse-mediated invasion of colon-26 and 4T1 cells inside the extracellular matrix of a collagen gel. We evaluated the invasiveness as the length of cell extension from the cancer cell cluster. This phenotypic change is caused by the alteration of several intracellular events, which are not fully characterized. A relevant cellular event is invasiveness as shown elsewhere [44]. It is expected that HS in the basement membrane surrounding the cell cluster is degraded during the invasion process. HS on the cell surface is also possibly involved in the invasion [4]. It is required to clarify what HS substrates are actually degraded during the invasion process in future studies. Regarding mesenchymal transition, we observed different features for an epithelial-to-mesenchymal transition during the cell extension experiment, that is, the upregulation of E-cadherin or downmodulation of N-cadherin was not observed. A matrix metalloproteinase MMP-2 was typically upregulated (data not shown). Expression of adhesion molecules has not been examined at present. Different assembly of collagen fibers in the presence of anionic molecules such as sulfated HA has been reported [48,49,50]. However, this is not likely the case. In the present study, hs-HA at a low concentration (27.8 µg/mL) was effective in the collagen gel with an estimated concentration of 3 mg/mL. A mixture of hs-HA and type I collagen at a ratio of 1:100 did not influence the fibrillogenesis of collagen I [48].

Chemokine and chemokine receptor systems are greatly involved in the metastasis process of colon cancer [51,52]. CCL2 or CCL2-CCR2 axis is relevant in recruiting tumor-associated macrophages to the site of cancer. Expression of CCL2 was reported to be a predictive marker for liver metastasis [53], and a factor to determine local metastasis [54] of human CRC. Sulfated HA effectively inhibited CCL2 release from colon-26 cells stimulated by exogenously added muHpse. In a previous study, we concluded that the stimulation did not require enzymatic activity of Hpse, and that cell surface HS is involved in the signal transduction of colon-26 cells [14]. In the present study, we confirmed that sulfated GAGs inhibited the binding of Hpse to the cell surface of colon-26 cells by using a cell ELISA-based method (Figure 5B). The next question is how, after binding of Hpse to the cell surface, the Hpse-mediated signal is intracellularly transduced. In the literature, Hpse-mediated binding stimulates macrophages via TLR-2 and -4 and subsequent NF-κB signaling [11]. We could detect the activation of an NF-κB-dependent signal in colon-26 cells using a luciferase reporter gene expression assay and confirmed that this signal is blocked or suppressed in the presence of sulfated HA. Although a possible link between cell surface HS and Toll-like receptors is expected as has been shown in previous studies [55,56], it is still unclear how HS-dependent binding of Hpse is transduced into an NF-κB-dependent signal. As suppression of CCL2 expression by sulfated HA would be beneficial to possibly decrease the accumulation of TAM, further studies are required to clarify molecular events of the sulfated HA-mediated suppression of CCL2 release.

The question arises whether there is a different, or common if any, manner of interaction between heparin and sulfated HA. Both are inhibitors of Hpse and possess basically similar anionic moieties, but they differ in their sulfation patterns and glycosidic linkages. Several carbohydrate-based inhibitors of Hpse, such as SST0001, PG545, a synthetic HS glycopeptide, and sulfated glycopolymers, have been modeled with huHpse based on the information of a crystallographic study [57,58,59,60]. FMO calculations provide both the quality and quantity of the molecular interaction energy at the atomic level. For the quality of the interaction, inter-fragment interaction energy (IFIE) and its energy components electrostatic (ES), exchange repulsion (EX), charge transfer and higher-order mixed term (CT+mix), and dispersion (DI) [61], are quantitatively evaluated (Appendix A). To correct the overestimation of ES interactions based on the negative charge of GAGs, a statistical correction of IFIE as SCIFIE was performed to evaluate the distant ES interactions. Here, the SCIFIE index was calculated to evaluate the binding properties of Hpse with hs-HA4 and heparin-derived Dp4.

A crystallographic study showed that the bound Dp4 chain is bent inside a binding cleft of the Hpse molecule, which facilitates close interaction of heparin with the enzyme cleavage site composed of negatively charged Glu225 and Glu343 that are potentially repulsive to sulfated carbohydrates [42]. Based on the molecular modeling and subsequent FMO calculation in the current study, it is predicted that the hs-HA4 chain that binds to huHpse has a relatively more extended structure than the Dp4 chain (Figure 6C), which makes it difficult for hs-HA4 to access the enzyme cleavage site. This is likely due to the difference in the glycoside linkage of the tetrasaccharides. Finding a basis for the molecular bending of Dp4 may possibly improve the efficacy of carbohydrate-based Hpse inhibitors by facilitating a closer attachment to the cleavage site of Hpse.

The SCIFIE of hs-HA4 was similar or slightly higher than that of Dp4. In both tetrasaccharides, the SCIFIE at (+1) and (−2) sugar residues was smaller, suggesting strong involvement of sulfated GlcNAc residues in the interactions (Appendix A). At each amino acid residue of Hpse, hs-HA4 and Dp4 showed similar SCIFIE patterns (Figure 7A,B), suggesting that the overall interactions of the tetrasaccharides with Hpse were similar. There are mainly two clusters of basic amino acid residues that are potentially involved in the substrate recognition of Hpse: heparin binding domain (HBD)-1 interacting with the non-reducing end of the substrate including Lys158, Lys159, and Lys161, and HBD-2 interacting with the reducing end including Arg272, Arg273, Lys274, Lys277 and Lys280 [62,63]. Both tetrasaccharides commonly interact with these amino acid residues in HBD-1 and HBD-2, as well as other amino acid residues Lys98, Lys231, Lys232, Arg303, etc. These basic amino acid residues located along the binding cleft of the Hpse molecule interact with the sulfated GAGs in an electrostatic manner. A crystallographic study indicated that *N*-sulfation in Dp4 is involved in the heparin-Hpse interaction, such as the 2-*N*-sulfate group in (−2) sugar interacting with Gly389 and Asn64, as well as the 2-*N*-sulfate group in (+1) sugar interacting with Lys231 and Arg303 [42]. Although hs-HA4 lacks 2-*N*-sulfation, other sulfate groups in the hs-HA4 can interact with Lys231 and Arg303, suggesting that Hpse is flexible for recognition of sulfation patterns on the GAGs. Another carbohydrate-based inhibitor of Hpse, PG545, has been studied using a combination of NMR experiments and molecular modeling [58]. The authors provide at least three possible binding patterns as a result of a docking study, which is different from hs-HA4 and Dp4.

The difference in SCIFIE of the two tetrasaccharides was further checked. The SCIFIE was lower in hs-HA4 at basic amino acid residues Lys98, Lys159, Lys161, Lys232, and Arg272 (Figure 7C and 7D). Lys159 and Lys161 are included in HBD-1, Arg272 in HBD-2. The SCIFIE between each sugar residue and the two HBDs is shown in Appendix A. In both heparin binding domains, binding to hs-HA4 was more stable than to Dp4. In hs-HA4, the (−2) sugar residue is greatly involved in the stronger interaction with HBD-1 (difference: −124.2 kcal/mol). The (−1) sugar residue (difference: −34.6 kcal/mol) is also involved. The (+1) and (−3) sugar residues (difference: −26.5 and −13.8 kcal/mol, respectively) are involved in the stronger interaction with HBD-2. Lys98 is located near HBD-1 (Appendix A). Lys158 whose information is missing in the basal data of the Protein Data Bank (PDBID:5E9C) possibly influences this interaction. Therefore, the relevance of the site-specific interaction between hs-HA4 and HBD-1 can be further emphasized. It is also noteworthy that HBD-1-derived peptide physically interacts with AGA*IA heparin-derived pentasaccharide (asterisk indicates a trisulfated saccharide) [62].

Molecular modeling of sulfated GAGs implies that three sulfate groups in hs-HA4 (4-*O*- and 6-*O*-sulfation in (−2) sugar residue and 2-*O*-sulfation in (−3) sugar residue) are relatively adjacent to each other (Figure 6A, arrows), whereas those of Dp4 (2-*N*- and 6-*O*-sulfation in (−2) sugar residue and 2-*O*-sulfation in (−3) sugar residue) are dispersed (Figure 6B). As a result, the 6-*O*-sulfate group of the (−2) sugar residue is in closer contact with Lys98 in hs-HA4 (1.92 Å) than in Dp4 (3.90 Å) (Appendix A), which leads to a stronger electrostatic interaction with Hpse and, consequently, efficient inhibition of enzymatic activity at a lower concentration (Figure 1) and a smaller dissociation equilibrium constant (Figure 3). Extending the knowledge on Hpse inhibition by different types of sulfated carbohydrates will be helpful to optimize the carbohydrate structure suitable for a Hpse inhibitor. Screening of materials that strongly interact with HBD-1 is also a promising strategy to identify potential Hpse inhibitors.

Sulfated HA has been administrated by intradermal [27,32], subcutaneous [33], intraperitoneal [35,46], or intracecal [38] injection depending on the disease sites, that is, delivery routes of sulfated HA are restricted. At present, sulfated HA with a molecular weight of 2000 has been examined [35]. To achieve better delivery efficiency of sulfated HA, we need to optimize the chemical structure of sulfated HA with smaller molecular weight and more hydrophobic.

## 4. Materials and Methods

### 4.1. Reagents and Cells

Recombinant proteins that mimic mouse proheparanase (muHpse(pro)) and mature Hpse (muHpse(mature)) were collected as supernatants of *Trichoplusia ni* insect cells and purified as described elsewhere [64]. The 8-kDa and 50-kDa genes of human Hpse (huHpse(mature)) were amplified and subcloned into pFastBac Dual (Thermo Ficher Scientific, Carlsbad, CA, USA) similar to muHpse [64]. The supernatant of virus-infected Sf9 cells was collected and the heterodimer recombinant protein was partially purified using heparin Sepharose with linear gradient elution from 0.5 to 1 M of NaCl. Separately, huHpse was partially purified from the lysates of 293T cells that stably expressed huHpse transgene using heparin Sepharose. Heparin from porcine intestinal mucosa was purchased from Sigma (H7005 and H3149, approx. 15 kDa, St. Louis, MO, USA). Sodium hyaluronate (NaHA-T2: 20~30 kDa) was purchased from PG Research (Tokyo, Japan). High and low-sulfated HA has been donated from Tokyo Chemical Industry, Co. Ltd. (Tokyo, Japan, H1739: 40~50 kDa and H1740: 50~60 kDa, respectively). Cell-matrix type I-A (631-00651, 3 mg/mL) from FUJIFILM Wako Pure Chemicals (Tokyo, Japan), BAY11-7082 from Adipogen (San Diego, CA, USA). Biotin-LC-hydrazide (B-3770) was purchased from Sigma and biotin-PEG4-hydrazide (B5578) from Tokyo Chemical Industry. Murine colorectal carcinoma colon-26 cells were kindly donated from Prof. Takashi Tsuruo (The University of Tokyo). Murine mammary carcinoma 4T1 cells (CRL-2539) and human embryonic kidney 293T cells (CRL-3216) were purchased from ATCC. Insect cells Sf9 and Tn (Trichoplusia ni) were obtained from Novartis Pharma Tsukuba Research Institute. Colon-26 cells, colon-26 A1-3 cells expressing muHpse transgene, and 4T1 cells were maintained as described elsewhere [14].

### 4.2. Hpse-Mediated HS Degradation

HS degradation was detected using Superdex™ 75 Increase (5/150 GL, Cytiva, Uppsala, Sweden) as described elsewhere [65]. To test the inhibitory effect of GAGs, Hpse samples were preincubated with different concentrations of GAGs for 30 min before the degradation test. The magnitude of the degradation was quantified from HPLC chromatograms by measuring an area of 0.25 < K_av_ < 1 of the elution and dividing it by the total elution area. Representative gel filtration results are shown in Appendix A.

### 4.3. Binding of Human and Murine Hpse to Immobilized GAGs

Streptavidin solution (10 µg/mL, 15 µL) was added to a half-size ELISA plate (Greiner 675061) for immobilization. After blocking with 1% BSA solution, various concentrations of GAGs, biotinylated on their carboxyl groups using biotin-LC-hydrazide or biotin-PEG4-hydrazide (20 µL), were added for immobilization. After washing, Hpse was added to the wells and incubated for 2 h. After washing, the plate was incubated with rabbit anti-Hpse antibody (1/1200 dilution) followed by horseradish peroxidase-conjugated goat anti-rabbit IgG antibody (1/2500 dilution). The binding of Hpse was quantified by color development with ABTS and absorbance measurement at 405 nm.

### 4.4. Surface Plasmon Resonance

The interaction of Hpse with heparin and hs-HA was examined using a BIAcore T200 system. The immobilization of biotinylated GAGs was confirmed by the observation of a ~28 resonance unit increase on a streptavidin-immobilized sensor chip (BR-1005-31, Cytiva, Uppsala, Sweden). The binding reactions were carried out at 20 °C. MuHpse(mature) dissolved in running buffer (HBS-EP+: 10 mM HEPES, 0.15 M NaCl, 3 mM EDTA, and 0.05% (*w*/*v*) P20, pH 7.3) was injected onto the sensor chips at a flow rate of 30 µL/min. Association and dissociation times were 120 s and 180 s, respectively. Kinetic parameters were evaluated with BIAevalution software 4.1 (Cytiva, Uppsala, Sweden) using a 1:1 binding model. Association and dissociation rate constants (k_a_ and k_d_) as well as dissociation equilibrium constants (K_D_) were determined.

### 4.5. Invasion Assay of 3D-Cultured Cancer Cell Clusters in a Collagen Gel

Cells were suspended in a serum-free medium at a density of 2 × 10^4^ cells/mL, and the cell suspension (4 mL) was added to an agarose gel with an embossed surface. After culturing for 2 d, the suspended cells formed cancer cell clusters in small “caves” formed on the agarose gel. The cell clusters (colon 26 A1-3, average diameter: 144 ± 24 µm (mean ± S.D.), *n* = 189; 4T1, average diameter: 153 ± 24 µm (mean ± S.D.), *n* = 211) were suspended in a chilled collagen solution on ice. In some experiments, GAGs were mixed with the collagen solution. The mixture (50 µL) was transferred to a 96-well plate. After complete gelation, 50 µL of the culture medium was added to the gel. Then the cell suspension was cultured for a further 48 h. The cell viability was maintained during the experimental process, as confirmed by the WST8 assay. Dendrite-like cell extensions from the cluster body were quantitatively evaluated, that is, the longest extension of cells away from the cell cluster was picked up and its length was measured. Data were shown as average and S.D. of the lengths derived from 11 to 21 cell clusters for each experimental condition.

### 4.6. Hpse-Dependent CCL2 Release from Colon-26 Cells as Well as Cellular Binding of Colon-26 Cells to Hpse

Colon-26 cells were cultured at a density of 2 × 10^4^ cells per well in a 96-well plate. After incubation for 24 h, the cells were washed and cultured in an RPMI1640 medium with 0.1% BSA with or without 400 ng/mL muHpse(pro) for 24 h. To examine GAG-mediated inhibition, muHpse(pro) was pretreated with GAG (200 µg/mL) for 30 min. To examine the effect of BAY 11-7082, the cells were pretreated with inhibitor solution (10 µM) for 30 min, then muHpse(pro) was directly added into the wells. During the experimental process, cell viability was always over 99% on a 96-well plate. The CCL2 concentration in the cultured supernatant was determined by sandwich ELISA (Murine JE ELISA Development Kit 900-K126, Peprotech, Rocky Hill, NJ, USA).

For detection of muHpse binding to the cell monolayer, the cells were similarly treated with 400 ng/mL muHpse(pro) in the 96-well plate. The plate was kept at 4 °C for 30 min, washed with a cold medium twice, and fixed with 4% paraformaldehyde solution for 10 min. After further washing, the wells were treated with 1% BSA solution in PBS, rabbit anti-Hpse antiserum (1/1200 dilution), and diluted horseradish peroxidase-conjugated goat anti-rabbit IgG (H + L) (1/2500 dilution). muHpse binding was quantified by color development with ABTS and absorbance measurement at 405 nm.

### 4.7. NF-κB-Dependent Luciferase Assay in Colon-26 Cells

A DNA construct for NF-κB response element, minimal promoter, and Nanoluc PEST (NlucP) reporter genes were amplified by PCR using pNL3.2 NF-κB-RE vector (Promega) as a template. PCR was conducted with the following set of primers: 5′- AGA TCC AGT TTA TCG ATG GGA ATT TCC GGG GAC TT -3′ and 5′- ATT GGA TCC GCG GCC GCT TAG ACG TTG ATG CGA GC -3′. The product was cloned into the *Cla I* and *Not I* restriction sites of pLVSIN-CMV Pur vector (Takara Bio) to establish pLVSIN-NF-κB/Nluc using In-Fusion HD cloning system. Lentiviral particles were produced in HEK293FT cells after the co-transfection of pLVSIN-NF-κB/Nluc and ViraPower (three packaging plasmids; Invitrogen; Carlsbad, CA, USA) using polyethylenimine “MAX” (Polysciences, Inc.; Warrington, PA, USA) and concentrated by the PEG precipitation method as described previously [66,67]. The virus solution was mixed with hexadimethrine bromide (8 μg/mL at final concentration), and the mixture was added to colon-26 cells (1 × 10^5^ cells in a 6-well plate). The plate was centrifuged (1800× *g*, 90 min) at 32 °C and incubated at 37 °C for 24 h. The cells were cultured for 48 h after replacement with RPMI 1640 medium supplemented with 10% FCS, followed by continuous selection with puromycin (2.5 μg/mL) to establish colon-26 NF-κB-Nluc cells.

Nano-Glo^®^ Luciferase Assay (Promega) was conducted as follows. Colon-26 NF-κB-Nluc cells were placed into a 96-well plate, cultured for 24 h, and washed with a serum-deprived medium. Fifty µL of a 400 ng/mL muHpse(pro) solution diluted in RPMI1640 medium containing 0.1% BSA was added. After 12 h of incubation, 50 µL of Nano-Glo^®^ Luciferase Assay Reagent was added and mixed. The mixed solution was transferred to the measurement plate and luciferase-mediated luminescence was measured using an ARVO X2 (Perkin Elmer, Waltham, MA, USA).

### 4.8. Statistical Analysis

Differences between treatment groups were evaluated using the two-tailed Student’s *t*-test. A *p*-value of 0.05 was considered statistically significant.

### 4.9. Fragmented Molecular Orbital Calculation

The X-ray crystal structure of the complex formed between Hpse and an inhibitor, Dp4, was downloaded from the Protein Data Bank (PDBID:5E9C). Modeling was performed in three steps in the following order; firstly addition of hydrogens and missing atoms in the ligand and amino acid residues using the integrated calculation chemistry system MOE [68] (Protonate 3D, pH = 7.0), secondly structural optimization of added hydrogen atoms using Amber10:EHT force field, and thirdly structural optimization of all atoms (Restraint:Tether = 1.0).

The Hpse protein contained a total of 434 residues (A-chain: K179-K538, B-chain: Q36-E109), four sugar chains, and 220 crystal water molecules. In the Dp4 complex, the sugar chains included in the X-ray crystal structure were ∆HexUA(2S)-GlcNS(6S)-IdoUA-GlcNS(6S), which is the minor component, and the hydroxyl group at the 2-*N*-position of IdoUA was replaced with a sulfate group to form the major component, ∆HexUA(2S)-GlcNS(6S)-IdoUA(2S)-GlcNS(6S). In the complex with hs-HA4, the 1,4-glycosidic linkage between the (−2) and the (−3) sugars was rearranged to a 1,3-glycosidic linkage and the functional group was replaced, as shown in Appendix A. The number of functional groups in both models was six sulfate groups and two carboxyl groups per tetrasaccharide, and the overall charge of the sugar was −8*e*.

FMO calculation was performed under the control of ABINIT-MP program using the supercomputer Fugaku at the level of FMO2-MP2/6-31G* [69]. For fragmentation, the protein part was divided into amino acid residues [40] and the glycan part into monosaccharide units (Appendix A) [70]. The Hpse-ligand binding moiety was analyzed based on the interaction energy (IFIE, PIEDA [61], and the calculated FMO was statistically modified using the SCIFIE [45]).

## 5. Conclusions

We demonstrated that sulfated HA, especially hs-HA, blocks Hpse-mediated enzymatic actions and cellular functions, that is, invasion into the surrounding extracellular matrix and Hpse-mediated upregulation of CCL2 released from colon-26 carcinoma cells. These findings have implications for the novel utilization of sulfated HAs as potentially anti-metastatic and anti-inflammatory agents via inhibition of Hpse functions.

## Figures and Tables

**Figure 1 ijms-23-05055-f001:**
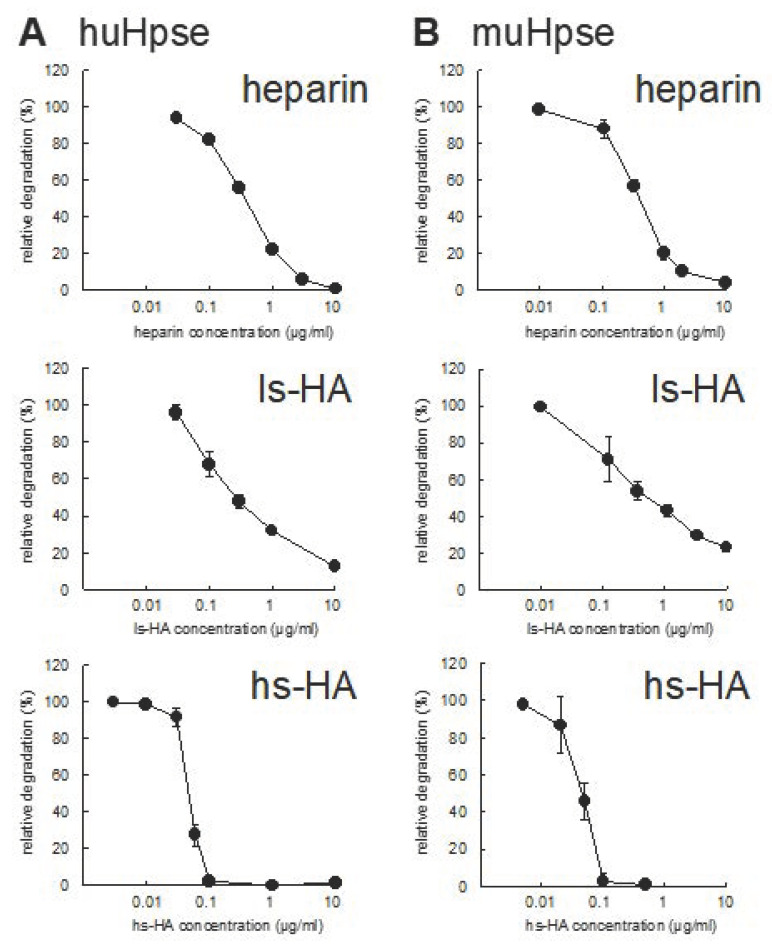
Heparan sulfate degradation by human (huHpse) or mouse (muHpse) heparanase was inhibited by sulfated glycosaminoglycans. (**A**) Partially purified huHpse and (**B**) recombinant muHpse(mature) (400 ng/mL) were pretreated with different concentrations of heparin, low-sulfated hyaluronan (ls-HA), and high-sulfated hyaluronan (hs-HA) for 30 min. Fluorescine-labeled heparan sulfate (Fl-HS) was added to the mixture and further incubated for 20 h. Degradation of Fl-HS was detected by gel filtration and plotted as relative degradation. Degradation by partially purified huHpse or 400 ng/mL of muHpse(mature) without any inhibitor was set to 100%. Data are shown as mean ± S.D. *n* = 3.

**Figure 2 ijms-23-05055-f002:**
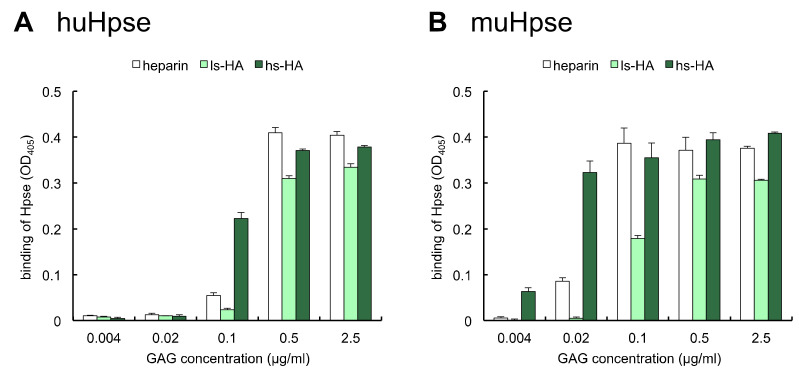
Human or mouse Hpse bound to immobilized sulfated GAGs. Different amounts of biotinylated sulfated GAGs were immobilized. Partially purified huHpse (**A**) or recombinant muHpse(mature) (**B**) were added, and Hpse bound to the immobilized sulfated GAG was detected by 2,2′-azino-bis [3-ethylbenzothiazoline-6-sulfonic acid] (ABTS) absorbance at 405 nm. Data are shown as mean ± S.D. *n* = 3.

**Figure 3 ijms-23-05055-f003:**
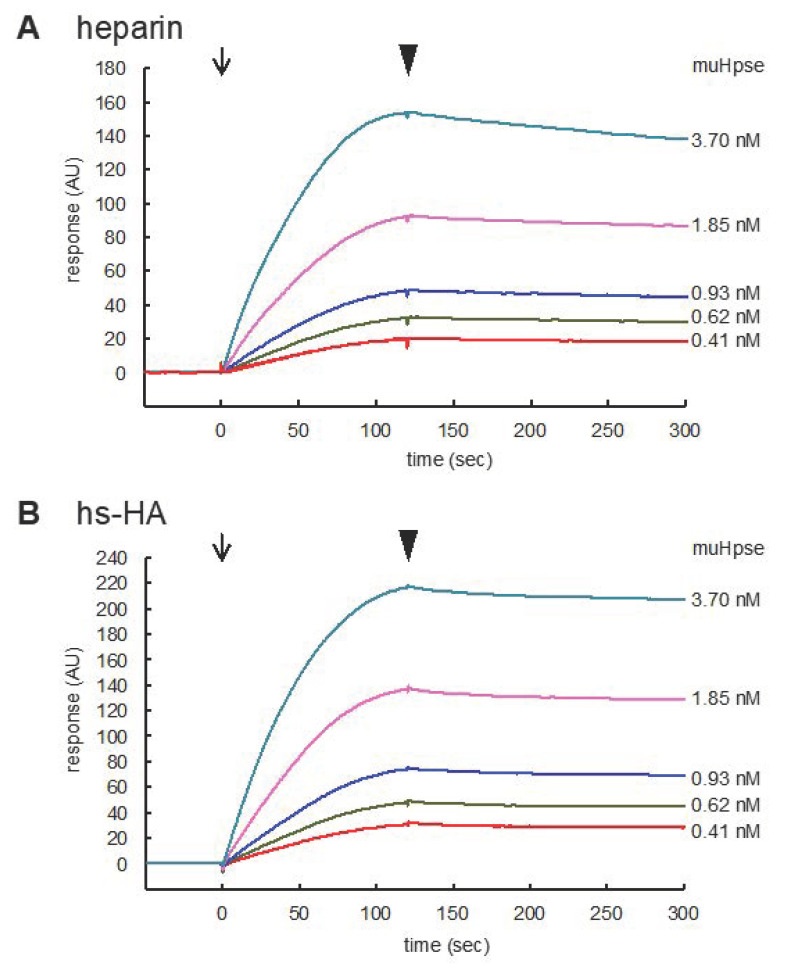
Sensorgrams for the binding of muHpse(mature) to immobilized sulfated GAGs. Recombinant muHpse(mature) (0.41–3.70 nM) was injected onto the surface of the sensor chips immobilized with biotinylated heparin (**A**) and hs-HA (**B**). The flow rate was 30 µL/min. The beginning of association and dissociation are indicated by arrows and arrowheads, respectively, in the figures. The sensorgrams were overlaid using BIAevaluation software.

**Figure 4 ijms-23-05055-f004:**
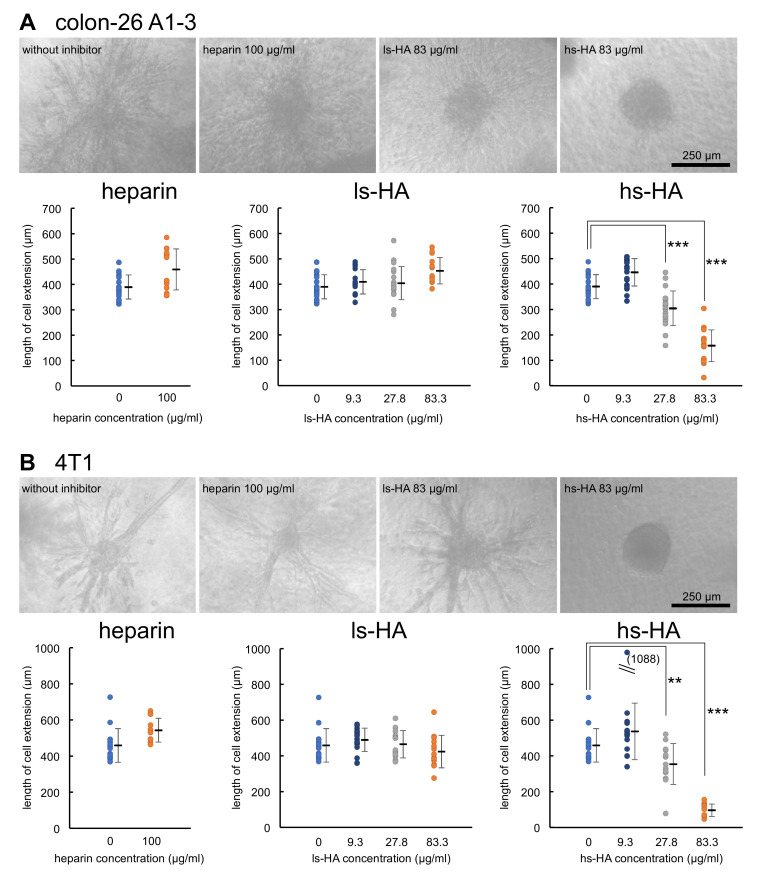
Cell extension of carcinoma cells in a collagen gel matrix was inhibited by the addition of hs-HA. Cell clusters derived from colon-26 A1-3 cells expressing muHpse transgene (**A**) and 4T1 cells (**B**) were embedded in collagen gels with or without addition of sulfated GAGs. After culturing for 48 h, the length of the longest extension was measured. (upper panels) Dendritic cell extension from the cell cluster, bar: 250 µm; (lower panels) The length of the longest cell extension was plotted. Data are shown as mean ± S.D. **, ***: Significantly lower than control (**: *p* < 0.01, ***: *p* < 0.001).

**Figure 5 ijms-23-05055-f005:**
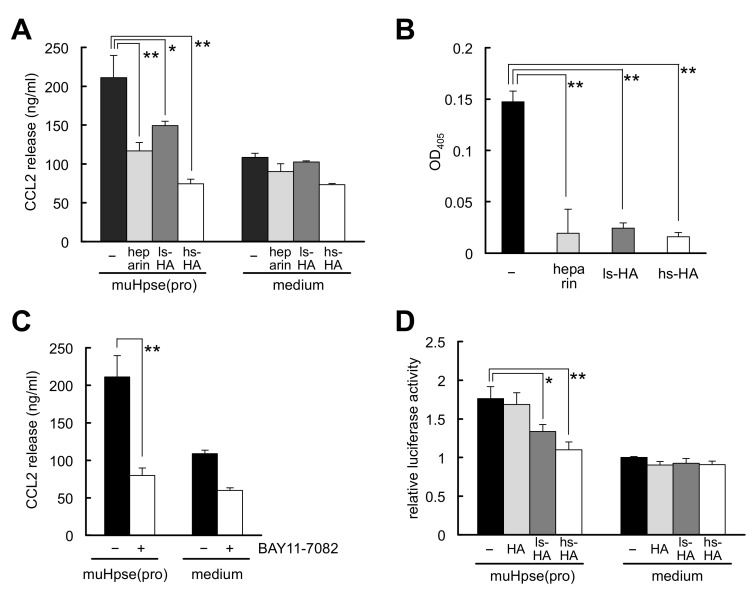
Hpse-mediated chemokine CCL2 release from stimulated colon-26 carcinoma cells was inhibited by sulfated GAGs. (**A**) muHpse(pro) was pretreated with sulfated GAGs (200 µg/mL) for 30 min and added to colon-26 cells cultured in RPMI1640 medium containing 0.1% BSA (final concentration of muHpse(pro): 400 ng/mL). The supernatant was collected after incubation for 24 h and the concentration of CCL2 was measured. (**B**) muHpse(pro) (400 ng/mL) was pretreated with sulfated GAGs (200 µg/mL) for 30 min, and added to colon-26 cell monolayer at 4 °C. After washing and fixation, bound muHpse(pro) was detected using a cell ELISA method. Absorbance at 405 nm was plotted after normalization to the absorbance of colon-26 cell monolayer that was not treated with muHpse(pro). (**C**) muHpse(pro) (400 ng/mL)-mediated CCL2 release was examined with cells that were pretreated with 10 µM of BAY 11-7082. (**D**) Luciferase expression of colon-26 cells expressing NF-κB RE vector. muHpse(pro) (400 ng/mL) was pretreated with HA, ls-HA or hs-HA (200 µg/mL) for 30 min and added to the cells. Luminescence was measured after incubation for 12 h. The luminescent value is shown as relative luciferase activity of untreated cells. Data are shown as mean ± S.D. *n* = 3 (**A**–**C**) or *n* = 4 (**D**). *, **: Significantly lower than control (*: *p* < 0.05, **: *p* < 0.01).

**Figure 6 ijms-23-05055-f006:**
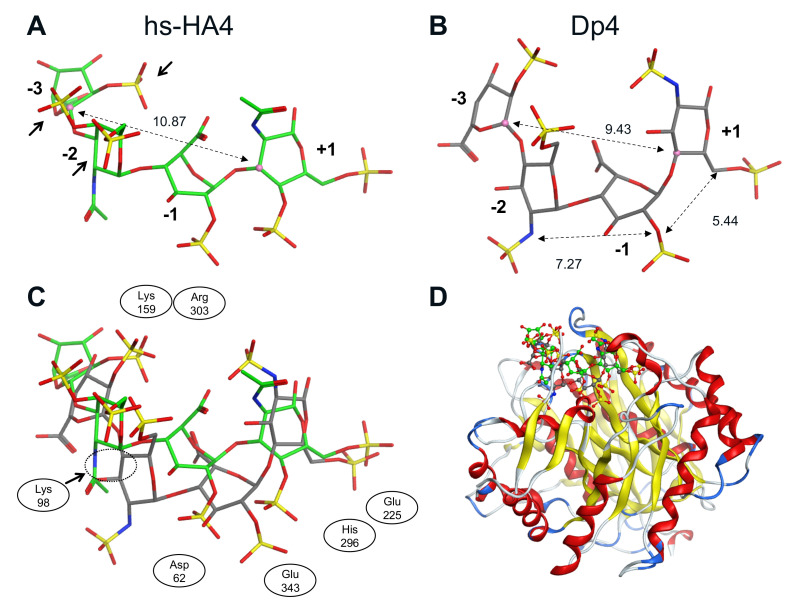
Conformation of sulfated GAG tetrasaccharides bound to a human Hpse molecule as a result of molecular modeling. (**A**) Conformation of hs-HA4 and (**B**) Dp4. Dotted lines indicate the distance between two sugar residues. The distance from (+1) to (−3) sugar was 10.87 Å in hs-HA4, whereas it was 9.43 Å in Dp4. Relatively adjacent sulfated groups in (−2) and (−3) sugar residues of hs-HA4 are indicated with arrows in panel A. (**C**) Superimposed conformations of hs-HA4 and Dp4. Adjacent amino acid residues are also indicated. (**D**) Predicted 3D complex of the tetrasaccharides with huHpse. Green and gray backbones indicate hs-HA4 and Dp4, respectively.

**Figure 7 ijms-23-05055-f007:**
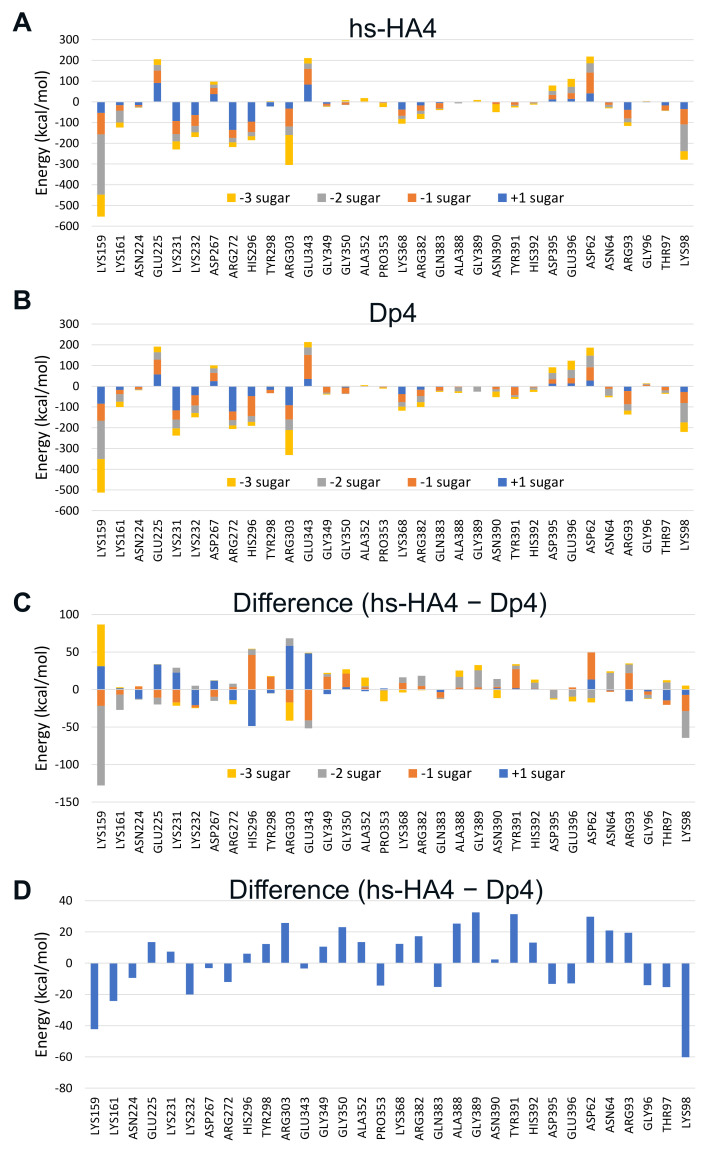
SCIFIE of sulfated GAG tetrasaccharides bound to a huHpse molecule. (**A**,**B**) SCIFIE of amino acid residues of huHpse bound to the tetrasaccharides. (**A**) hs-HA4, (**B**) Dp4. (**C**,**D**) Difference in SCIFIE in panels A and B, where the difference is shown for each sugar residue (**C**) or shown as the cumulative SCIFIE difference (**D**). Amino acid residues with a difference of more than 10 kcal/mol between the two tetrasaccharides are shown in the panel.

**Table 1 ijms-23-05055-t001:** IC_50_ values of sulfated glycosaminoglycans (GAGs) to inhibit enzyme activity of human (huHpse) and murine (muHpse) heparanase.

Sulfated GAGs	IC_50_ for HuHpse (µg/mL)	IC_50_ for MuHpse (µg/mL)
Heparin	0.363 ± 0.006	0.400 ± 0.028
ls-HA	0.267 ± 0.061	0.542 ± 0.236
hs-HA	0.047 ± 0.003 ***	0.046 ± 0.007 ***

Data are shown as mean *±* S.D., *n* = 3. *** Significantly smaller than heparin (*p* < 0.001). ls-HA; low-sulfated hyaluronan, hs-HA; high-sulfated hyaluronan.

**Table 2 ijms-23-05055-t002:** IC_50_ values of sulfated GAGs to inhibit enzyme activity of murine Hpse.

Sulfated GAGs	k_a_ (M^−1^s^−1^)	k_d_ (1/s^−1^)	K_D_ (M)
heparin	(1.18 ± 0.42) × 10^7^	(9.58 ± 0.58) × 10^−4^	(8.16 ± 0.72) × 10^−11^
hs-HA	(1.20 ± 0.18) × 10^7^	(3.08 ± 0.52) × 10^−4^ ***	(2.67 ± 0.85) × 10^−11^ ***

*** Significantly smaller than heparin (*p* < 0.001). Data are shown as mean ± S.D. *n* = 3.

## Data Availability

Structure files and a set of input/output files used for FMO calculations are available at the FMODB (https://drugdesign.riken.jp/FMODB/, accessed on 21 April 2022); FMODBID: 9G4Y2 (hs-HA4) and LJZL9 (Dp4). Simple data analysis can be performed using the FMODB web interface, and detailed analysis can be performed using the BioStation Viewer software (https://fmodd.jp/biostationviewer-dl/, accessed on 21 April 2022).

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
