# Peer review of "Sulfated Hyaluronan Binds to Heparanase and Blocks Its Enzymatic and Cellular Actions in Carcinoma Cells"

_ijms, 2022, doi:10.3390/ijms23095055_

Round 1
Reviewer 1 Report
In the current study, the authors evaluated sulfated-hyaluronan as a potential inhibitor of heparanase. While I am not an expert in Fragmented Molecular Orbital Calculations, the biochemistry and biology aspects of the current study are sound. The results support their hypothesis, and the paper is well-written overall.
As a minor point, how do the researchers foresee bypassing the limitations associated with sulfated glycosaminoglycans? For example, bleeding side-effects that were seen in PI-88 clinical trials or the difficulty with oral delivery? While highly sulfated long polymers might serve as good tool compounds, do the researchers see these findings translating to the clinic?
Reviewer 2 Report
The authors examined whether sulfated hyaluronan (shHA) exerts inhibitory effects on enzymatic and biological actions of heparanase. Since this enzyme participates in cancer invasion and malignancy heparanase is a cancer target and its inhibitors are considered in cancer therapy strategies. The authors conducted very well the experiments that demonstrate the physicochemical interaction between shHA and heparanase (human and murine). However, the experiments about the cellular actions of shHA associated with its capacity to inhibit heparanase are weak. Thus, the authors should consider some suggestions for publishing their results
First, they should check that the shHA effects are not associated with cellular death, such as apoptosis or necrosis, which also can reduce the extension of cell clusters in the 3D culture and chemokine expressions. So, they should analyze the effect on the cell’s survival in this experiment.
- In the invasion assay, they should rule out the shHA action over other enzymes able to degrade GAG and PGs, for example, hyaluronidases, which also are associated with the invasion process in tumor cells. Or what about the effects over surface PGs, like syndecan, that are cell surface heparan sulfate PGs.
- Regarding the anti-inflammatory effects, the authors analyzed the shHA action over the CCL2 expression. But there is no explanation why they selected studies on this chemokine. Is CCL2 a key molecule in the inflammatory state in colorectal cancer ?. They should review this revision : Pączek, S., Łukaszewicz-Zając, M., & Mroczko, B. (2020). Chemokines-What Is Their Role in Colorectal Cancer? Cancer control : journal of the Moffitt Cancer Center, 27(1), 1073274820903384. https://doi.org/10.1177/1073274820903384. And, they should explain the rationale for this selection beyond the citation of his previous work.
